# Altered Cerebral Curvature in Preterm Infants Is Associated with the Common Genetic Variation Related to Autism Spectrum Disorder and Lipid Metabolism

**DOI:** 10.3390/jcm11113135

**Published:** 2022-05-31

**Authors:** Hyuna Kim, Ja-Hye Ahn, Joo Young Lee, Yong Hun Jang, Young-Eun Kim, Johanna Inhyang Kim, Bung-Nyun Kim, Hyun Ju Lee

**Affiliations:** 1Department of Translational Medicine, Hanyang University Graduate School of Biomedical Science and Engineering, Seoul 04763, Korea; ruykaz1@hanmail.net (H.K.); ks1253@hanyang.ac.kr (J.Y.L.); ryanjang93@hanyang.ac.kr (Y.H.J.); 2Department of Pediatrics, Hanyang University College of Medicine, Seoul 04763, Korea; mdscully@gmail.com; 3Department of Laboratory Medicine, Hanyang University Hospital, Seoul 04763, Korea; young0eun@hanyang.ac.kr; 4Department of Psychiatry, Hanyang University Hospital, Hangyang University College of Medicine, Seoul 04763, Korea; iambabyvox@hanmail.net; 5Division of Child and Adolescent Psychiatry, Department of Psychiatry, Seoul National University Hospital, Seoul 03080, Korea

**Keywords:** preterm, imaging genetics, common variation, *OXTR*, *FADS2*, *COMT*, M-CRIB-S

## Abstract

Preterm births are often associated with neurodevelopmental impairment. In the critical developmental period of the fetal brain, preterm birth disrupts cortical maturation. Notably, preterm birth leads to alterations in the fronto-striatal and temporal lobes and the limbic region. Recent advances in MRI acquisition and analysis methods have revealed an integrated approach to the genetic influence on brain structure. Based on imaging studies, we hypothesized that the altered cortical structure observed after preterm birth is associated with common genetic variations. We found that the presence of the minor allele at rs1042778 in *OXTR* was associated with reduced curvature in the right medial orbitofrontal gyrus (*p* < 0.001). The presence of the minor allele at rs174576 in *FADS2* (*p* < 0.001) or rs740603 in *COMT* (*p* < 0.001) was related to reduced curvature in the left posterior cingulate gyrus. This study provides biological insight into altered cortical curvature at term-equivalent age, suggesting that the common genetic variations related to autism spectrum disorder (ASD) and lipid metabolism may mediate vulnerability to early cortical dysmaturation in preterm infants.

## 1. Introduction

Preterm delivery accounts for 11% of all births [1] and is associated with neurodevelopmental impairment [2]. Preterm infants exhibit abnormal structural and functional brain development, reflecting a high risk of long-term neurocognitive problems. Preterm birth can affect cortical development and is linked to vulnerability to neuropsychiatric disorders, characterized by cognitive deficits in later life [3].

Preterm birth disrupts cortical maturation during the critical period of fetal brain growth [4]. In normal development, the interaction between the faster tangential growth of the cortex and the underlying white-matter core contributes to local changes in the gyrification and sulcation patterns of the cerebral cortex [5]. The prenatal and postnatal periods of brain development, with an extensive period of synaptic modulation and pruning, are observed throughout the first year of life [6]. As very preterm infants are born during the course of cortical development, they are vulnerable to dysmaturation in specific cortical regions. To better understand the role of various factors in altered cortical development in preterm infants, it is important to understand the maturational features of the cortical index, such as volume [7], thickness [8], and curvature [9]. Compared with controls, the brain volumes of preterm infants displayed global reductions [10]. The thickness [8] and curvature [11,12] of the cortex are biomarkers of neurological health, and both increases and decreases are observed in individuals born very preterm. However, the studies reported different results, depending on the local area of the brain; therefore, local alterations to the cortex must be considered to identify preterm infants’ cerebral development.

Our previous study [13] proposed that preterm births were associated with altered lateralization of the fronto-limbic circuitry, reflecting deficits in functional specialization in specific regions rather than deficits in the global brain network. Notably, preterm birth leads to alterations in the fronto-striatal and temporal lobes, and the development of the limbic region takes place in a critical period of fetal brain development. The fronto-limbic pathway primarily consists of the orbitofrontal, the anterior cingulate, insula, and limbic regions, including the amygdala and hippocampus, which is known as the brain’s affective network, underlying the processing and regulation of emotions [14,15,16]. The impact of prematurity on cerebral development and damage has been extensively studied using magnetic resonance imaging (MRI) [17], which provides sensitive assessments of abnormal morphology and microstructure that correspond with neurodevelopmental outcomes [18,19]. Recent advances in MRI acquisition and analysis methods have opened possibilities for the diagnosis of neurological diseases and disorders based on abnormalities in cortical morphology, revealing an integrated approach to the genetic influence on brain structure [20]. In particular, the susceptibility to cerebral abnormalities is likely to be modulated by the combined effects of multiple genes of individual and environmental factors [21] during pregnancy and the early postnatal period [22]. Altered cerebral development in preterm infants is associated with environmental stress and cerebral endophenotypes [23]. An endophenotype, an intermediate phenotype between genotypes and phenotypes, is a quantitative trait [24] related to the root cause of a disease [25,26]. The molecular and cellular mechanisms involved are not well known, but new approaches have made it possible to investigate the relationship between common genetic influences and brain endophenotypes [27]. In molecular genetic studies, the use of brain structural measures as endophenotypic biomarkers would clarify the pathophysiology of major psychiatric disorders.

In our previous study, using nationwide birth cohort data, we found that preterm birth is a significant risk factor for autism spectrum disorder (ASD), which is characterized by impaired social communication and restricted/repetitive behavior [28,29]. However, important questions also remain regarding the contribution of genetic factors to the link between neurodevelopmental delay and preterm birth. Although spontaneous preterm birth is a multifactorial condition, an individual’s genetic variation affects susceptibility and prognosis in prematurity-related disease after preterm birth. The cortex is particularly involved in genetically driven processes, and the expression of specific growth factors, such as oxytocin receptor (*OXTR*) [30], fatty acid desaturase 2 (*FADS2*) [31], and catechol-o-methyltransferase (*COMT*) gene [32], plays a crucial role in driving cortical development. Therefore, the single-nucleotide polymorphisms (SNP) in the *OXTR*, *FADS2*, and *COMT* genes, which modulate lipid metabolism and the risk of cerebral abnormalities, might affect cortical development in preterm infants.

These genes were selected because of previous studies related to prefrontal brain function and social/behavioral dysfunction, which is linked to psychiatric disorders, including ASD [33,34,35] and schizophrenia [36,37,38]. We aimed to provide insight into the genetic influence on brain development to identify infants with an inherent vulnerability to reduced development of the cerebral cortex after preterm birth. Based on previous findings on the genetic impact on developmental processes in the cortex [30,31,32], we hypothesized that the altered cortical structure observed after preterm birth is associated with common genetic variations involved in ASD and lipid metabolism.

## 2. Materials and Methods

### 2.1. Study Design and Participants

In this study, we included only subjects of Korean descent to minimize the effects of ethnicity on population stratification. In accordance with the World Health Organization (WHO), we defined preterm birth as any birth before 37 complete weeks of gestation and subdivided them based on gestational age (GA): very preterm (28 -< 32 weeks) and late preterm (32 –< 37 complete weeks of gestation) [39,40]. This study included a routine brain MRI at term-equivalent age (TEA) for 14 very preterm infants (GA < 32 weeks) from the neonatal intensive care unit of Hanyang University Seoul Hospital. Since preterm infants are at higher risk of developing intracranial hemorrhage or ischemia, routine MRI was used as standard practice as part of a routine screening to detect acquired lesions associated with prematurity. The 29 late preterm infants (GA between 32 and 37 weeks) were recruited from a prospective observational cohort study involving postnatal follow-up of preterm infants who were admitted to the neonatal intensive care unit of Hanyang University Seoul Hospital, where brain MRI for late preterm TEA is a part of our follow-up program. All preterm infants (14 very preterm and 29 late preterm) visited the Hanyang Inclusive Clinic for Developmental Disorders in Hanyang University between August 2018 and December 2020. We excluded 13 preterm infants with major chromosomal anomalies (*n* = 0), congenital malformations (*n* = 0), and intrauterine growth restriction (IUGR) (*n* = 0), as well as those with evidence of visible brain injury (*n* = 4) and those whose imaging was of quality due to motion artifacts (*n* = 9). The selected infants were prospectively enrolled as part of the Medicine Engineering Bio Research Project at the Hanyang Inclusive Clinic for Developmental Disorders, and their parents consented to DNA collection. Six term-born neonates were selected from a prior prospective cohort of controls who underwent MRI in the neonatal period, recruited as part of preterm studies [13], and whose parents consented to the collection of DNA at Hanyang University Medical Center Pediatrics Department.

### 2.2. Genotyping

We collected genetic data for twenty-six infants (nine very preterm infants, <32 weeks; eleven preterm infants, 32–37 weeks; and six term-born infants, >37 weeks of gestational age). Single-nucleotide polymorphism (SNP) array analyses were performed on DNA samples from 26 subjects and divided into the preterm and control groups. Each group was divided into two groups based on the existence of the minor allele, allowing a comparison of minor homozygotes or heterozygotes with major allele homozygotes, and the allelic and genotypic frequencies of polymorphisms between the preterm and control groups were compared. Nine nucleotide polymorphisms from *OXTR*, *FADS2*, and *COMT* were used to assess the relationship between brain volume, thickness or curvature, and genotype.

Normalized bead-intensity data obtained for each sample were loaded into Beeline or AutoConvert s/w software (Illumina, San Diego, CA, USA), which converted fluorescence intensities into SNP genotypes. To identify and cluster genotypes, each SNP was analyzed independently. Comparing those in the supplied cluster file (*.egt) with experimental data, genotype calls were generated. Over 100 samples were used to create a standard cluster from the Caucasian (CEU), Asian (CHB + JPT), and Yoruban (YRI) HapMap populations, which should have incorporated most of the genetic diversity in these populations. In general, the genetic diversity of the samples was well represented by the standard cluster file. Thus, genotypes were automatically clustered using a standard cluster file. In some cases, the sample intensities might not have overlapped perfectly with the standard cluster position. In these cases, we reclustered some or all SNP to optimize the genotype calling. Genotype calls were extracted using the Illumina BeadArray Files library. Nine SNPs for *OXTR*, *FADS2*, and *COMT* were extracted for this study, and all SNPs significantly deviated from the Hardy–Weinberg equilibrium (Table 1).

### 2.3. MRI Scanning Procedure

According to the institutional neonatal MRI guidelines for scanning methods, we obtained MRI scans of all infants [19,41,42]. The timing of the MRI scans was scheduled according to the infants’ nap times, ranging from 20 to 30 min. We did not use a sedative drug because preterm newborns are at risk of respiratory instability, including physiological changes, equipment compatibility, and acoustic noise during MRI scanning. We used earplugs, headphones, or wrap techniques to circumvent these. The attending physician conducted a detailed review of the medical history and a physical examination. Every 5 min, the vital signs and cardiorespiratory conditions were recorded. Parents were informed about the MRI process at TEA.

### 2.4. MRI Acquisitions

The preterm and control groups underwent whole-body 3T MRI (Philips, Achieva, 16-channel phase-array head coil, Best, Netherlands) at near-term ages during natural sleep with a blanket to maintain body temperature. An experienced pediatrician monitored the pulse oximeter to ensure the respiration rate and heart rate during the MRI scan. The parameters of the T2 scan were as follows: slice numbers = 25, slice thickness = 3, voxel sizes = 0.5 × 0.5 mm^2^, field of view (FOV) = 180 × 180 mm^2^, repetition time (TR) = 4800 ms, time to echo (TE) = 90 ms, flip angle = 90 degrees, number of average = 1, water-fat shift = 4.68 Hz/pixel, and total acquisition time = 6 min 30 s. Among the acquired MRI data, only the area presented in Figure 1 was used for analysis.

### 2.5. Image Processing

A commonly used labeling atlas in children and adults is the Desikan–Killiany–Tourville (DKT) atlas [43]. However, compared to the adult brain, the tissue signal intensities of the cortical surfaces of neonates with rapid brain development show different MRI sequences according to age. We used Melbourne Children’s Regional Infant Brain Atlases (M-CRIB-S), which constructed a labeling atlas using T2-weighted images that provide higher tissue contrast in neonates, to examine the automatic segmentation of cortical and subcortical brain areas and extract cortical surfaces [44]. M-CRIB-S is useful for detecting regional alterations in gray matter to reveal the impact of genotype effects on the structural architecture of neural substrates for neurodevelopmental impairment after premature birth. The overall flowchart for image processing and reconstruction of the cortical surfaces is presented in Figure 2. To improve the low-frequency intensity inhomogeneity, bias filed estimation was performed from the T2-weighted images using improved N3 bias correction (N4ITK) [45]. The brain extraction tool (BET) was applied to remove the skull and nonbrain tissue. Each T2-weighted image was non-linearly registered to multiple pre-labeled images automatically using the Developing Brain Region Annotation with Expectation-Maximization (DrawEM) software package [46], and segmented into cerebral gray and white matter, cerebellum, and subcortical structures. The deformable module of Medical Image Registration ToolKit (MIRTK) (https://github.com/BioMedIA/MIRTK, accessed on 22 March 2022) was used to extract the outer and inner outlines of the cortical surfaces. FreeSurfer tools (https://surfer.nmr.mgh.harvard.edu, accessed on 22 March 2022) [47], which include linear and nonlinear registration, were used to reconstruct inflated and spherical surfaces. The inflated surfaces showed the same overall shape features as when FreeSurfer was run on adult brain images. The volumetric M-CRIB-S (DKT) labels were projected onto the corresponding vertices of the white-matter surfaces using the nearest labeled neighbor projection and parcelated on each T2-weighted image.

### 2.6. Neurodevelopmental Assessment

The Korean Developmental Screening Test for Newborns and Children (K-DST), created in September 2014 by the Korean Pediatric Society and presided over by the Korean Standard Classification of Diseases (KCD) [48], was used to examine the infants’ neurodevelopment. The K-DST was developed and modified to detect probable developmental delays by considering the culture and newborn care environment in Korea [49]. The K-DST consists of questionnaires for six domains (gross motor function, fine motor function, cognition, language, social interaction, and self-help). Because self-help ability emerges after a certain developmental stage, it is tested from the age of 18 months, and was not measured in this study. The total score of each domain was assigned to one of the following four: higher than peer-level (≥1 SD), peer-level (<1 SD and ≥−1 SD), follow-up evaluation is recommended (<−1 SD and ≥−2 SD), and detailed evaluation is warranted (<−2 SD).

### 2.7. Statistical Analysis

The t-tests on continuous variables and X^2^ tests on categorical variables were performed to determine the statistical significance of the differences between the preterm and control groups. We calculated global cortical thickness, average absolute mean curvature, and global cortical volume in the left and right hemispheres. We also measured the regional cortical thickness and mean curvature of the fronto-limbic pathway. We used the false discovery rate (FDR) correction for multiple comparisons. Analysis of covariance for both groups, adjusted for age at scan, was used to compare the genetic variables and cerebral structures.

## 3. Results

This study included 26 infants of both sexes, and the *OXTR*, *FADS2*, and *COMT* genes were genotyped in all of them. We compared the demographic characteristics of the preterm and control infants. All the infants in both groups were Asian and without brain injury. The mean gestational age of the preterm infants was 31.00 ± 3.88 weeks and that of the control infants was 38.83 ± 0.98 weeks. The rates of moderate-to-severe bronchopulmonary dysplasia (BPD) and retinopathy of prematurity (ROP), which is known as an independent risk factor for neurodevelopmental impairment, were 15% and 10%, respectively, in preterm infants. The clinical and demographic data are summarized in Table 2.

### 3.1. The Differences in Cerebral Volume, Thickness, and Curvature between the Preterm and Control Groups

We collected MRI data for the 26 infants (nine very preterm infants, <32 weeks; eleven preterm infants, 32–37 weeks; and six term-born infants, >37 weeks of gestational age). There were no significant differences in brain volume and thickness, but the preterm infants had a decreased mean cortical curvature in the right medial orbitofrontal gyrus (FDR-corrected *p* = 0.028), right superior temporal gyrus (FDR-corrected *p* = 0.010), left posterior cingulate gyrus (FDR-corrected *p* = 0.028), and left superior temporal gyrus (FDR-corrected *p* = 0.028) after correcting for age at scan (Table 3).

### 3.2. The Effect of Genotype on Cerebral Volume, Thickness, and Curvature

The presence of the minor allele at rs1042778 in *OXTR* was associated with reduced curvature in the right medial orbitofrontal gyrus (*p* < 0.001). The presence of the minor allele at rs174576 in *FADS2* (*p* < 0.001) or rs740603 in *COMT* (*p* < 0.001) was related to a reduced curvature in the left posterior cingulate gyrus (Figure 3, Appendix A). However, there was no significant relationship between the SNP genotype and cerebral volume or thickness (Appendix A). There was no significant difference between the minor allele frequencies of the SNPs and gestational age (Appendix A). The sex differences in cerebral volume, thickness, and curvature were not significant (Appendix A).

### 3.3. The Neurodevelopmental Follow-Up

At the 1-year follow-up in this study, all the participants selected had no major neuropsychiatric disease and no major health problems. The K-DST were compared between the preterm infants and the term-born infants, and there was no significant difference between the five specific domains and within at least one of the five domains (Table 2). The infants were considered screen positive for the K-DST if any of the six domains had a score of 2 SD or higher, indicating that the infant should be investigated further for suspected developmental delay. Two preterm infants received a score of <-2 SD in any domain, with one in the language domain and the other in the gross motor domain.

## 4. Discussion

We found that the carriage of the minor alleles in *OXTR*, *FADS2*, and *COMT* was associated with reduced curvature after controlling for age at scan. We observed that SNPs linked to the genetic regions associated with ASD and lipid metabolism predicted changes in the cerebral endophenotype.

In certain developmental and neuropsychiatric disorders, specific abnormalities are reflected in an individual’s developing cortex morphology and may correspond to pathological functioning. With the development of technology, studies are underway to reveal the relationship between the brain and genes in preterm infants [50,51]. In particular, preterm infants exhibit a more immature cortical microstructure and macrostructure, resulting in an atypical trajectory of cortical maturation at term-equivalent age [52]. Although the exact mechanism of altered cortical development after preterm birth is not clearly elucidated, previous studies proposed a genetic pathway that modulates preterm brain injury [53,54]. Many studies have demonstrated the inheritability of folding patterns [55,56], showing that curvature has a high degree of genetic determination.

The preterm infants exhibited reduced development in the cerebral cortex, including the cerebral volume and the complexity of cortical folding, which validates and expands previous published findings [10,12]. Preterm birth per se might influence delayed cortical maturation, followed by long-term neurocognitive impairment [18]. In addition, Kline et al. [11] showed that decreased cortical volume and curvature are also promising predictors of later neurodevelopment in preterm infants. However, no studies have examined the genetic mechanism of altered brain development at TEA in preterm infants compared to term-born infants. Boardman et al. [23] found that common genetic variation in the genes associated with schizophrenia modulates the susceptibility to white-matter abnormalities by using diffusion tensor imaging at TEA, but they only studied a preterm population.

We identified that the common genetic variation in the genes associated with neurodevelopmental delay affect cortical curvature significantly in preterm-born infants compared to term-born infants. Despite decades of study, the genetic mechanism behind the prematurity-related factors that affect complex folding patterns have not been fully elucidated in preterm infants. The clinical implication of our study is that cortical development in specific areas could differ significantly between preterm and term infants, depending on the carriage of particular alleles at SNP, which are risk factors for genetic susceptibility. Our study aims to elucidate that the combination of preterm birth and genetic factors might interfere with the regional complexity of cortical folding, even in the absence of major white-matter damage.

We found that minor allele carriage at rs1042778 in *OXTR* is a risk modulator for the reduced mean curvature of the right medial orbitofrontal gyrus in preterm infants. *OXTR*, located on chromosome 3p25, is a neuropeptide that is highly conserved in brain areas critical for emotional regulation, such as the amygdala, lateral septum, and hypothalamus. A SNP of *OXTR* has recently emerged as a particularly interesting candidate gene for social–behavioral phenotypes [30,57]. Notably, rs1042778 plays a crucial regulatory role in *OXTR* transcription and translation. A common genetic variant in *OXTR* linked to social function predicts individual differences in the hypothalamic–limbic structures of healthy adults [58]. Other studies have reported that preterm birth is associated with cortical volume [59] and neuronal loss in the dorsomedial nucleus of the thalamus [60]. The prefrontal cortex, which exerts top-down control over various subcortical regions, densely expresses oxytocin receptors that may mediate the effects of oxytocin on social behavior. Common genetic variants in *OXTR* were associated with the behavioral traits of ASD, including deficits in social communication and cognition, restricted and repetitive behaviors [33,61], and emotional empathy, in a Chinese cohort [62]. Despite accumulating evidence supporting the relationship between preterm birth and ASD, the intermediate neural mechanism in preterm infants remains unknown. In this study, the *OXTR* gene was associated with the curvature of the orbitofrontal gyrus, which is involved in the cognitive process of decision-making and socio-emotional functioning.

The presence of a minor allele at rs174576 in *FADS2* and at rs740603 in *COMT* was associated with lower curvature of the left posterior cingulate. The posterior cingulate gyrus is called the limbic lobe and plays a prominent role in cognitive functions, such as working memory performance [63,64]. It is one of the most metabolically active regions of the brain and has been linked to spatial memory by lesion studies, as well as to emotional salience [65]. Several neurological and psychiatric disorders are associated with abnormalities in this area, including Alzheimer’s disease, ASD, ADHD, depression, and schizophrenia [64]. Our results suggest that the *FADS2* and *COMT* gene is associated with posterior cingulate gyrus curvature, which might raise the possibility of genetic vulnerability in the preterm phenotype.

*FADS2* polymorphisms modify the activity of long-chain polyunsaturated fatty acid (LC-PUFA) desaturation and lipid composition in brain tissue, modulating cortical development in children. LC-PUFAs are important for neurogenesis, neurotransmission, and protection against oxidative stress because they accumulate in the brain during the last trimester of pregnancy. Genetic variations in *FADS2* may interact with preterm brain maturation to influence later neurodevelopment. Our results are similar to recently reported data by Boardman et al. [23] showing that the carriage of the minor allele at rs174576 in the *FADS2* gene is linked to preterm white-matter injury after preterm birth, based on diffusion tensor imaging [31,66]. However, to our knowledge, there has been little genetic evidence on cortical curvature in preterm neuroimaging studies to suggest the reduced complexity of the preterm brain compared to the term-born brain. Our results indicate the genetic impact on the developmental process in the maturation of the cortex in the absence of white-matter injury in preterm infants. The *FADS2* gene is associated with brain-related phenotypes, such as intellectual development [67] and attention-deficit hyperactivity disorder (ADHD) [68]. The *COMT* gene is a key enzyme in the metabolism of dopamine, norepinephrine, and epinephrine [69,70] and has been associated with cognitive functions [71]. In addition, the SNPs in *COMT* are associated with white-matter changes in preterm-born adults [32] and have a potential impact on dopamine regulation and the phenotypic traits of ASD patients [34]. This shows that *COMT* may trigger difficulties in premature metabolism and cognitive development in newborns. The important limitations of these data merit consideration. Despite our positive findings, our study featured a small sample size. Additionally, the results of the present study must be interpreted cautiously and corroborated by independent and multicenter studies to determine the actual prevalence of *OXTR*, *FADS2*, and *COMT* polymorphisms and their association with preterm morbidity. Furthermore, the M-CRIB atlas that we used is based on a dataset of normal term-born neonates; therefore, it may not map precisely onto the brain morphology of infants with plagiocephaly.

## 5. Conclusions

In conclusion, we identified that the altered cortical structure in preterm infants compared to term-born infants is associated with common genetic variations involved in ASD and lipid metabolism. Significantly reduced curvature of the fronto-limbic pathway at TEA was found in the carriage of the minor alleles in *OXTR*, *FADS2*, and *COMT*. This study provides biological insight into altered cortical curvature at term-equivalent age, suggesting that common genetic variations related to ASD and lipid metabolism may mediate vulnerability to early cortical dysmaturation in preterm infants. Our imaging-based biomarkers of underlying genetic predispositions might offer promise for identifying relevant genetic variations and pertinent imaging features. With larger sample sizes, more findings are likely to emerge, which makes it possible to solve non-genetic confounders in future studies. Since we explored structural brain development, future studies need to analyze areas of functional development, such as brain networks.

## Figures and Tables

**Figure 1 jcm-11-03135-f001:**
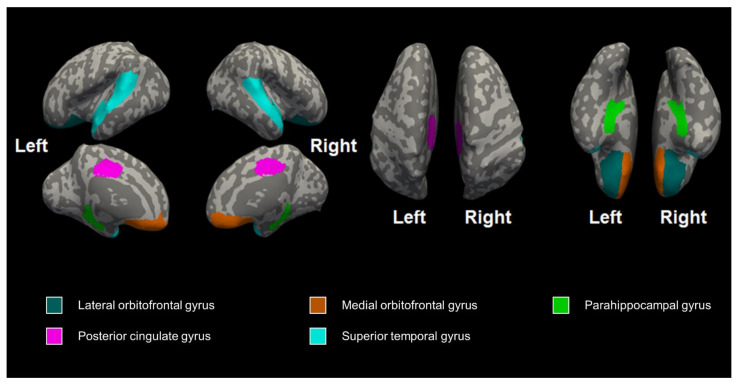
The regions of the brain curvature that we observed.

**Figure 2 jcm-11-03135-f002:**
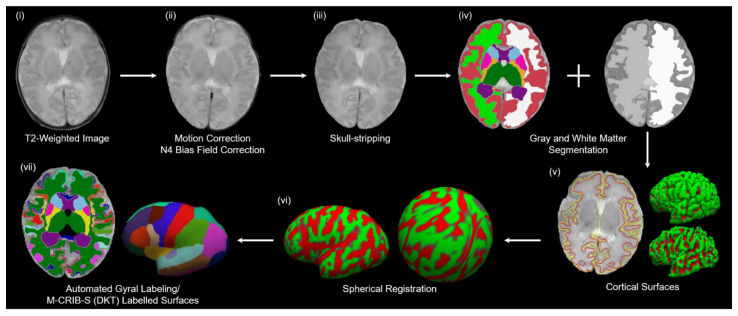
Flowchart of image processing. (**i**) The original T2-weighted image; (**ii**,**iii**) preprocessing of the T2-weighted image; (**iv**) automatically segmented regions (cerebral gray and white matter, cerebellum, and subcortical gray matter); (**v**) extracted cortical surfaces. The left panel exhibits outer (yellow), and inner (red) cortical surfaces overlaid onto the native image. The right-top and right-bottom panels show a lateral view of the left hemisphere outer and inner surfaces; (**vi**) inflated and the spherical surfaces of the white-matter surfaces; (**vii**) automated gyral parcelation using the M-CRIB-S neonate atlas.

**Figure 3 jcm-11-03135-f003:**
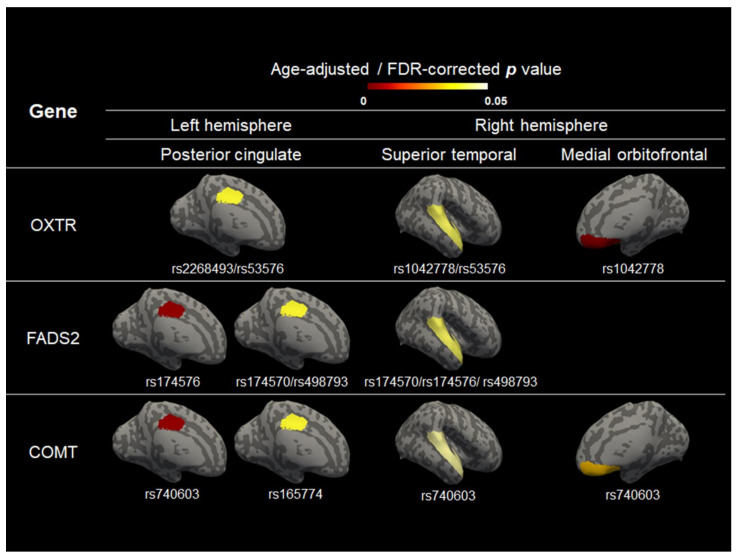
The significant regions of the brain curvature associated with OXTR, FADS2, and COMT.

**Table 1 jcm-11-03135-t001:** Allele frequencies, coding information, linkage disequilibrium details, and location of SNP.

Gene	SNP Name	Chromosome	Coordinate (Position)	Source	Variant	East Asian MAF	Hardy–Weinberg Equilibrium Unaffected, *P*	Gene Location
*OXTR*	rs1042778	3	8794545	dbSNP 153	G/T	0.093	1.000	3′UTR
*OXTR*	rs2268493	3	8800840	dbSNP 153	T/C	0.210	1.000	Intron
*OXTR*	rs53576	3	8804371	dbSNP 153	A/G	0.419	1.000	Intron
*FADS2*	rs174570	11	61597212	dbSNP 153	C/T	0.387	1.000	Intron
*FADS2*	rs174576	11	61603510	dbSNP 153	C/A	0.556	0.857	Intron
*FADS2*	rs498793	11	61624705	dbSNP 153	C/T	0.107	1.000	Intron
*COMT*	rs740603	22	19945177	dbSNP 153	A/G	0.426	1.000	Intron
*COMT*	rs165774	22	19952561	dbSNP 153	G/A	0.149	0.797	Intron
*COMT*	rs174696	22	19953176	dbSNP 153	C/T	0.422	1.000	Intron

SNP, single-nucleotide polymorphism; *OXTR*, oxytocin receptor; *FADS2*, fatty acid desaturase 2; *COMT*, catechol-o-methyltransferase.

**Table 2 jcm-11-03135-t002:** Descriptive characteristics of the cohort.

Characteristics	Preterm (*n* = 20)	Control (*n* = 6)	*p*-Value
Sex			0.182
Male	13 (65.0%)	2 (33.3%)	
Female	7 (35.0%)	4 (66.7%)	
Ethnicity			
Asian	20 (100%)	6 (100%)	NA
Scan age, mean ± SD	37.60 ± 2.04	40.00 ± 1.79	0.016
Birth weight, mean ± SD	1624.00 ± 710.40	3373.33 ± 27.53	<0.001
Gestational age, mean ± SD	31.00 ± 3.88	38.83 ± 0.98	<0.001
Moderate-to-severe BPD	3 (15%)	0 (0%)	0.412
Moderate-to-severe ROP	2 (10%)	0 (0%)	0.420
K-DST			
<-2 SD in any domain	2 (10%)	0 (0%)	0.509
<-2 SD in gross motor domain	1 (5%)	0 (0%)	0.648
<-2 SD in fine motor domain	0 (0%)	0 (0%)	NA
<-2 SD in cognition domain	0 (0%)	0 (0%)	NA
<-2 SD in language domain	1 (5%)	0 (0%)	0.648
<-2 SD in sociality domain	0 (0%)	0 (0%)	NA

SD, standard deviation; NA, not available; BPD, bronchopulmonary dysplasia; ROP, retinopathy of prematurity; K-DST, Korean developmental assessment of infants.

**Table 3 jcm-11-03135-t003:** Brain differences between preterm and control infants.

Characteristics	Very Preterm (*n* = 9)	Late Preterm (*n* = 11)	Control (*n* = 6)	*p*-Value	FDR
Volume					
Total gray-matter volume	98,268.70 ± 11,511.49	109,480.18 ± 15,158.94	116,044.91 ± 10,982.82	0.043	0.072
Left hemisphere cortical gray matter	34,697.14 ± 4374.11	36,820.41 ± 5824.50	38,987.16 ± 5452.27	0.317	0.317
Right hemisphere cortical gray matter	32,924.48 ± 3688.58	36,505.99 ± 5754.01	38,354.64 ± 3806.95	0.091	0.114
Left-hemisphere cerebral white matter	63,135.95 ± 7821.90	73,906.42 ± 10,090.29	72,867.42 ± 9780.54	0.040	0.072
Right-hemisphere cerebral white matter	60,970.21 ± 8171.73	70,194.39 ± 7638.09	69,560.52 ± 6089.01	0.027	0.072
Thickness					
Left lateral orbitofrontal gyrus	1.08 ± 0.15	1.15 ± 0.20	1.15 ± 0.23	0.661	0.826
Right lateral orbitofrontal gyrus	1.03 ± 0.12	1.11 ± 0.21	1.09 ± 0.12	0.522	0.826
Left medial orbitofrontal gyrus	1.14 ± 0.17	1.16 ± 0.15	1.18 ± 0.16	0.877	0.937
Right medial orbitofrontal gyrus	1.19 ± 0.23	1.22 ± 0.14	1.12 ± 0.14	0.574	0.826
Left parahippocampal gyrus	1.06 ± 0.22	1.01 ± 0.14	1.10 ± 0.07	0.584	0.826
Right parahippocampal gyrus	0.98 ± 0.25	1.07 ± 0.19	1.03 ± 0.14	0.588	0.826
Left posterior cingulate gyrus	1.34 ± 0.19	1.33 ± 0.12	1.31 ± 0.07	0.937	0.937
Right posterior cingulate gyrus	1.30 ± 0.15	1.36 ± 0.08	1.36 ± 0.13	0.551	0.826
Left superior temporal gyrus	1.41 ± 0.17	1.38 ± 0.13	1.45 ± 0.16	0.645	0.826
Right superior temporal gyrus	1.34 ± 0.14	1.46 ± 0.17	1.42 ± 0.14	0.256	0.826
Curvature					
Left lateral orbitofrontal gyrus	1.76 ± 0.74	1.96 ± 0.51	2.47 ± 0.30	0.075	0.107
Right lateral orbitofrontal gyrus	1.73 ± 0.46	2.23 ± 0.58	2.43 ± 0.70	0.064	0.107
Left medial orbitofrontal gyrus	1.35 ± 0.63	1.63 ± 0.31	2.00 ± 0.38	0.045	0.090
Right medial orbitofrontal gyrus	1.20 ± 0.40	1.79 ± 0.40	1.86 ± 0.59	0.011	0.028
Left parahippocampal gyrus	0.55 ± 0.16	0.52 ± 0.31	0.62 ± 0.31	0.735	0.735
Right parahippocampal gyrus	0.35 ± 0.16	0.49 ± 0.52	0.55 ± 0.27	0.571	0.634
Left posterior cingulate gyrus	0.54 ± 0.27	1.01 ± 0.30	1.00 ± 0.40	0.006	0.028
Right posterior cingulate gyrus	0.76 ± 0.41	0.98 ± 0.29	0.89 ± 0.23	0.324	0.405
Left superior temporal gyrus	4.41 ± 0.81	5.35 ± 0.62	5.69 ± 0.99	0.009	0.028
Right superior temporal gyrus	3.70 ± 0.62	4.64 ± 0.83	5.32 ± 0.57	<0.001	0.010

FDR, false discovery rate.

## Data Availability

Not applicable.

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
