# Peer review of "Altered Cerebral Curvature in Preterm Infants Is Associated with the Common Genetic Variation Related to Autism Spectrum Disorder and Lipid Metabolism"

_jcm, 2022, doi:10.3390/jcm11113135_

Round 1
Reviewer 1 Report
This study investigates brain curvature alterations in infants formerly born premature. Furthermore, genetic aspects were considered. The authors elaborate on supratentorial curvature in a modest sample of 26 subjects. However, there are important limitations that need consideration by the authors. Furthermore, although the authors obviously made an effort in English-scientific-writing, this manuscript needs proofreading by an English native speaker. There are some phrases that do not properly convey the intended meaning and need streamlining by an expert in English-writing prior to publication. Find my suggestions below:
Title:
- Do not use abbreviations in the title without defining them – fully write out (and/or define) ASD in the title
Abstract:
- Define the abbreviation ASD in the abstract
Introduction:
- The introduction is missing a paragraph that describes why specific genetic aspects were focused. Although there is some information in this regard provided in the methods section, the authors should re-arrange the corresponding elements of this article and specify/characterize the genes of interest in the introduction.
- Line 58: Do the authors refer to the fronto-limbic pathway instead of “front-limbic pathway”? Please keep the error rate to a minimum prior to submission!
- Provide a final paragraph at the end of the introduction that clearly states what the aims of this study were and how the authors addressed these. E.g., This study aimed to … . For this purpose … .
Methods:
- Indicate whether this study was designed retrospectively or prospectively
- Line 86 – 89: “To assess the results of genetic association studies, all participants should have the same ethnic and geographical origins because of the genetic basis of the diseases. The influence of ethnic and geographic diversity can be extensive and should be considered in a genetic study”. The authors should skip this sentence.
- Information about specific (e.g., neuropsychiatric) diagnoses of the subjects included should be given.
- The authors should indicate at least the most common referral reasons for MRI in the neonatal period (for both preterm and term-born neonates).
- Line 102/103: “nine very preterm infants, <32 weeks; 11 preterm infants, 32–37 weeks; and six term-born infants, >37 weeks of gestational age” – provide respective references for these assertions, since there are various definitions of different prematurity states.
- The authors should provide a (supplementary) table that describes the whole MRI protocol used and its technical features due to reproducibility issues.
- The authors should provide more information on the post-processing procedure. Which steps (in which post-processing tool) were conducted to obtain the information of interest?
Results:
- Are there any gene-related characteristics in infants born at term with regard to volume, thickness and curvature? However, the control cohort of n=6 is rather modest to reliably elaborate on this topic.
Discussion:
- The authors should consider the fact that gyration, sulcation, and opercularization (and in a broader sense brain curvature) in former preterm born infants (imaged at term-equivalent ages) is generally delayed (as compared to term-born infants). Thus, it appears reasonable that there are differences in brain curvature between former preterm and term-born infants. The authors should dedicate a paragraph of the discussion to this issue and justify why their results withstand this important limitation (which was not mentioned by the authors in the limitations section).
- Did the authors consider whether the included infants revealed plagiocephaly? In the exclusion criteria in the methods section and the limitations section there is no comment on this, although positional plagiocephaly could have an impact on the provided data.
- The authors should clearly state the limitations of this study. The small sample size is not a “potential limitation” – the small sample is an important limitation, since there are only 6 subjects in the control group. Important aspects that might have had an impact on the provided data are not mentioned (see above). Information about any (neuropsychiatric, but also other) diagnoses across the sample size is not given. There is evidence that brain maturation differs in females and males formerly born extremely premature – this study involves males and females. Thus, the authors should provide a statement in this regard in their limitations section that takes potential sex-related differences into account.
Figures:
- Adequate
Tables:
- Table 2 is missing a section with regard to potential diagnoses across the cohort.
Supplementary Material:
- Adequate
Author Response
Dear. Reveiwer 1
Thank you for the thoughtful comments, which helped us substantial improvement the paper. We made changes to the manuscript and tables based on your feedback. For your convenience, the revisions are mentioned with line no. for each of your point-by-point comments. Additionally, the major changes we made are marked in the revised manuscript.
Please see the attachment file.
Kind regard

Reviewer 2 Report
Dear Ahn et al.,
The manuscript “Altered Cerebral Curvature in Preterm Infants Is Associated with the Common Genetic Variation Related to ASD and Lipid Metabolism” (jcm-1671106) by Ahn et al. provides biological insight into the altered cortical curvature at term-equivalent age, reflecting that common genetic variation related to ASD, and lipid metabolism may mediate vulnerability to early cortical dysmaturation in preterm infants. Some of my specific comments are below:
- In the abstract section (line 17-28), the authors should add quantitative results rather than only qualitative results.
- In the keywords section (line 29-30), the keywords is too much and recommend to reducing it.
- Describe the novelty of the article made by the author? From the results of my evaluation, it seems that many similar published works adequately explain what you have raised in the current manuscript related to Cerebral Curvature on Autism Spectrum Disorder and Metabolism based on the best reviewer knowledge in this research area. If there are something others really new in this manuscript, please highlight it more clearly in the introduction section (line 32-83).
- The state of the art and the significance of the current study are not clearly present, the authors should highlight it more advanced in the introduction section (line 32-83).
- In the introduction section (line 32-83), the authors should explain the previous research conducted and its shortcomings. For example A et al., conduct ……., their finding ….., their shortcomings. It will uphold the research gap that you filled with your research novelty. I recommend the authors elaborate on their introduction section. Do not forget to attention carefully to my previous comments on numbers 3 and 4.
- Since this manuscript related to autism spectrum disorder, I would encourage and advise the authors to adopt some of the specific additional reference related to autism spectrum disorder in the introduction section (line 32-83) as follow:
-
- Physiological Effect of Deep Pressure in Reducing Anxiety of Children with ASD during Traveling: A Public Transportation Setting. Bioengineering 2022, 9, 157. https://doi.org/10.3390/bioengineering9040157
- In the methods section (line 84-167), the authors should add one systematic figure to illustrate the workflow of experimental testing in the present study to make the reader more interested and easier to understand rather than only using dominant text to explain.
- For participants in the current study (line 85-100), are there standardizations or baselines? because the number of participants and participant criteria will greatly affect the results. This increasingly needs to be highlighted considering the small number of participants which makes the data from the research not become strong and leads to misinterpretation of the results. Sampling criteria for participants also need to be further detailed.
- In the Results section (line 168-196), the authors are advised to compare the results they obtain with previous similar/identical studies if it is possible.
- The conclusion (line 272-275) of the present manuscript is not solid. Further elaboration is needed.
- Further research needs to be explained in the conclusion section (line 272-275).
I am pleased to have been able to review the author's present manuscript. Hopefully, the author can revise the current manuscript as well as possible so that it becomes even better. Good luck for the author's work and effort.
Best regards,
The Reviewer
Author Response
Daer. Reviewer 2
Thank you for the thoughtful comments, which helped us substantial improvement the paper. We made changes to the manuscript and tables based on your feedback. For your convenience, the revisions are mentioned with line no. for each of your point-by-point comments. Additionally, the major changes we made are marked in blue in the revised manuscript.
Please see the attachment file.
Kind regard

Round 2
Reviewer 1 Report
The authors made an effort to improve this manuscript! The points raised by the reviewers have been addressed adequately.
Reviewer 2 Report
Dear Ahm et al.,
After carefully reading the author's revised manuscript entitled "Altered Cerebral Curvature in Preterm Infants Is Associated with the Common Genetic Variation Related to ASD and Lipid Metabolism" (jcm-1671106) by Ahm et al., The authors have been made significant improvements in the revised manuscript. Also, all of the issues in my review report have been addressed precisely.
With my pleasure, I recommend the manuscript should be accepted for publication on Journal of Clinical Medicine.
Best regards,
The Reviewer